# Regulation of inflammatory genes in decidual cells: Involvement of the bromodomain and extra-terminal family proteins

**Sandeep Ajgaonkar[1], Jonathan J. Hirst[1,2], Mary Norris[1,3], Tamas Zakar[1,2,3] ***

**1** College of Health, Medicine, and Wellbeing, University of Newcastle, Callaghan, NSW, Australia,
**2** Mothers and Babies Research Program, Hunter Medical Research Institute, New Lambton Heights, NSW,
Australia, **3** Department of Maternity and Gynaecology, John Hunter Hospital, New Lambton Heights, NSW,
Australia

* Tamas.Zakar@newcastle.edu.au

pone.0280645

Torino, ITALY

**Data Availability Statement:** The minimal dataset
underlying the results described in the paper can
be found in the Supporting Information.

## Abstract

The decidua undergoes proinflammatory activation in late pregnancy, promoting labor. Bro-
modomain and Extra-Terminal (BET) family proteins interact with acetylated histones and
may control gene expression in inflammation. Here, we assessed whether BETs are
involved in inflammatory gene regulation in human decidual cells. We have treated primary
cultures of decidual stromal cells (DSCs) from term pregnancies with endotoxin (LPS) and
measured the expression of a panel of pro-and anti-inflammatory genes. BET involvement
was assessed using the selective BET inhibitors (+)-JQ1 and I-BET-762 or the negative
control compound (-)-JQ1. Histone 3 and -4 acetylation and BETs binding at the target gene
promoters were determined to assess whether these processes are involved in the actions
of LPS, BETs, and BET inhibitors. LPS increased the expression of the proinflammatory
(*PTGS2*, *IL6*, *CXCL8/IL8*, *TNF*) and the anti-inflammatory (*IL10*, *IDO1*) genes of the panel.
The constitutively expressed inflammatory genes (*PTGS1*, *PTGES*) were unaffected. The
BET inhibitors, but not the control compound, reduced the basal and LPS-induced expres-
sion of *PTGS1*, *PTGS2*, *IL6*, *CXCL8/IL8*, *IL10*, and *IDO1*. *TNF* expression was not changed
by BET inhibition. The dominant BETs were Bromodomain-containing protein -2 (BRD2)
and -4L (BRD4L) in DSCs. LPS increased histone 4 acetylation at the *CXCL8/IL8* and *TNF*
promoters and histone 3 and -4 acetylation at the *IDO1* promoter, while (+)-JQ1 abrogated
histone acetylation at several promoters. Overall, histone acetylation and promoter binding
of BETs showed no consistent relationship with gene expression across the gene panel and
the treatments. BET proteins, predominantly BRD2 and BRD4L, control critical pro- and
anti-inflammatory genes in DSCs. *TNF* induction exemplifies a BET-independent pathway.
Changing histone acetylation at the promoters is not a general obligatory requirement for
inflammatory gene expression in response to LPS. BETs likely act at chromatin loci sepa-
rate from the examined promoters. BET inhibitors may block decidual activation at labor.

**Funding:** Funding for the study was received from the Hunter New England Local Health District, Grant Numbers: Charitable Trust Grant Round 2020 and 2021 (T.Z, M.N.) Faculty of Health and Medicine, University of Newcastle, Australia, Research Training Scheme (S.A.) The funders had no role in study design, data collection and analysis, decision to publish, or preparation of the manuscript.

**Competing interests:** The authors have declared that no competing interests exist.

# Introduction

The decidua is the endometrium of pregnancy, forming the maternal side of the maternal-fetal interface. It plays a crucial role in protecting the semi-allogenic fetus from immune rejection while preserving decidual competence to elicit an immune response against infections ascending from the vagina or originating from peripheral sources. These contrasting roles are performed by tightly controlled pro- and anti-inflammatory, immunosuppressive and immune-promoting responses [1, 2]. Disturbances of this balance are associated with pregnancy disorders such as miscarriage, pre-eclampsia, early membrane rupture, and premature labor.

The decidua is a complex tissue with diverse and dynamic cellular composition. It contains a variety of bone marrow-derived CD45[+] leukocytes that include monocytes/macrophages, dendritic cells, granulocytes, NK (natural killer) cells, T and B lymphocytes, and mast cells that change in abundance during gestation [3]. The largest proportion of cells is a resident decidual cell type establishing the stroma of the tissue. Decidual stromal cells (DSCs) exhibit pro- and anti-inflammatory capacities and influence leukocytes locally to adapt to the idiosyncratic needs of pregnancy [1]. DSCs promote Th2 immune bias [4], inhibit natural killer (NK) cell cytotoxicity, and reduce the activity of antigen-presenting (dendritic) cells [5] at the maternal-fetal interface favoring pregnancy maintenance. The pregnancy-protective phenotype of the decidua vanes towards the end of gestation [6, 7], and decidual cells shift to a proinflammatory phenotype [8], which promotes myometrial activation and cervical remodeling. This change is part of the process leading to labor and birth. Pathological activation of the decidua leads to dysfunctional labor, untimely delivery, and newborn complications. The mechanisms that underlie the inflammatory transformation of the decidua in normal or pathological labor are poorly understood. In culture, DSCs respond to LPS (lipopolysaccharide, an inflammatory trigger from Gram-negative bacteria) or cytokines like IL-1 beta and TNF-a by producing a variety of proinflammatory factors, including IL-6, IL-8, PTGS2, PTGES [9–13]. Anti-inflammatory cytokine production (IL-10) has also been reported after LPS treatment [14], indicating a balanced pro-and anti-inflammatory response.

Inflammatory transformation of cells involves widespread gene expression changes associated with alterations of chromatin structure at promoters, enhancers, and super-enhancers controlling inflammatory genes [15, 16]. The altered acetylation state of histones and non-histone proteins has been recognized as a critical component of the process [17]. One initial event is the binding of BRD4, (bromodomain-containing protein 4) a member of the BET (bromodomain and extra-terminal) family of proteins, to gene regulatory regions in a complex with NFkB (nuclear factor NF-kappa-B), the central regulator of inflammation [18]. BET recruitment is critical for activating super-enhancers and promoters controlling proinflammatory genes [19, 20], while BETs dissociating from other super-enhancers deactivates non-inflammatory genes during the phenotype transition [21]. BET family proteins bind acetyl-lysine residues in acetylated histone and non-histone proteins through their tandem bromodomains and protein interaction domains [22], which is the mechanistic basis of their chromatin interactions. Importantly, BET proteins can be targeted effectively by drugs that disrupt the bromodomain-acetyl lysine binding [23, 24], which abrogates the super-enhancer mediated cell phenotype transitions [19]. BET proteins are emerging central players in inflammation with modifiable activity [25].

The role of BET proteins in inflammatory gene regulation in the decidua is undefined. Here we have addressed this gap of knowledge by determining (1) the effect of BET inhibitors on LPS-induced inflammatory gene expression in primary cultures of DSCs, (2) the expression of BET family members in DSCs, and (3) the effects of LPS and BET inhibitor on histone acetylation and BET protein binding at the promoters of inflammatory genes in the DSCs. The

results demonstrate that BETs regulate pro- and anti-inflammatory genes in the decidua, and BET inhibitors abrogate the expression of the LPS-induced pro- and anti-inflammatory genes. The results also suggest that BRD2 and BRD4, the dominant BET proteins in DSCs, interact predominantly with genomic regulatory elements distant from the promoters of the inflammatory genes studied. Nonetheless, BET inhibitors can potentially block decidual inflammation with therapeutic benefits.

# Materials and methods

## Ethics approval

The study has been approved by the Hunter New England Human Research Ethics Committee (Project ID: 2019/ETH12332 and 2019/ETH12330). All participants donated placental tissue following informed written consent.

## Isolation, purification, and culture of decidual stromal cells

**Tissue processing.** Decidua tissue was isolated from placentae delivered by elective Cesarean section (in the absence of labor) at term after uncomplicated singleton pregnancies. None of the donors were diagnosed with intrauterine infection, inflammation, or histological chorioamnionitis. The reflected membranes were cut off at approximately 2 cm from the placental disk. The amnion was peeled away, and decidua tissue was scraped off the maternal side with a glass microscope slide. The scrapings (12-15g from one placenta) were washed in DPBS (Dulbecco's phosphate-buffered saline containing 40 μg/mL Gentamicin sulfate) and suspended in 25 mL digestion buffer (DMEM/F12 containing 20mM HEPES, pH7.4; 60 μg/mL Gentamicin sulfate; and the following enzymes: collagenase, Worthington LK003240, 1500 units; Dispase II, Sigma D4693, 62.5 mg, and DNAse I, Sigma DN25, 1.25 mg. Digestion was performed at 37˚C in a shaker water bath until the tissue was disintegrated entirely, which took 2–3 hours. The digest was centrifuged at 800 g for 10 min at room temperature, and the sediment was suspended in 25 mL of DMEM/F12 supplemented with 20 mM HEPES, pH7.4, 20% (v/v) heat-inactivated FBS, and 40 μg/mL Gentamicin sulfate. After centrifugation (600 g, 10 min, room temperature), the pellet was suspended in 10 mL of serum-free DMEM/F12 with 20 mM HEPES, pH 7.4, and 40 μg/mL Gentamicin sulfate, passed through a 120 μm pore size strainer and processed for Percoll density purification.

**Percoll purification.** "100% Percoll" was prepared by combining nine parts (vol) of the commercially supplied Percoll suspension (Sigma P1644) with one part of 10 x Hank's salt solution (Ca$^{++}$ and Mg$^{++}$-free) 200 mM HEPES, pH 7.4 and 40 μg/mL Gentamicin sulfate. Discontinuous density gradients of 50% overlaid with 30% Percoll were prepared, and the cell suspension, adjusted to 25% Percoll, was layered on the top. The gradients were centrifuged at 900 g for 40 min at room temperature. Cells were recovered from the 50%/30% boundary and washed to remove residual Percoll.

**Immuno-magnetic enrichment of DSCs.** Enrichment was achieved using the *EasySep* Human CD45$^+$ cell depletion kit (II) from StemCell Technologies (Tullamarine, VIC Australia), following the manufacturer's instructions. Briefly, cells were suspended at $10^8$ cells/mL concentration in Hank's salt solution supplemented with 20 mM HEPES, pH 7.4, 2% (v/v) FBS, and 40 μg/mL Gentamicin sulfate. A Depletion Cocktail containing anti-CD45 antibodies was added, and CD45$^+$ cells (mostly leukocytes [26]) were absorbed by adding magnetic RapidSpheres™ to the suspension. RapidSpheres™ with absorbed cells were removed using a strong magnet, and the enriched DSCs were suspended in cell culture medium (DMEM/F12, 20% FBS, 10 nM estradiol, 100 nM medroxyprogesterone acetate, 40 μg/mL Gentamicin sulfate). Cell yields were typically between 4.5–5.5x$10^7$ with 80 to 85% viability (Trypan blue

exclusion). The absence of CD45$^+$ cells was verified by flow cytometry in preliminary experiments. A representative example of the flow cytometry analysis of freshly isolated decidual cells is presented in S1 Fig, showing that >99% of live single cells were CD45 negative after purification.

**Cell culture.** Cells were seeded at $10^5/cm^2$ in cell culture flasks with gelatin (Type-2, Sigma G1393)—coated culture surface. Cultures were incubated in a humidified atmosphere of 5% $CO_2$ in air at 37˚C. The medium was changed every third day until 10–14 days of culture. Immunocytochemical characterization of the sub-confluent cultures (S2 Fig) showed pervasive expression of vimentin, a marker of stromal cells in the decidua [26], and the absence of CD45$^+$ cell contamination. The cultured DSCs were treated with (+)-JQ1, I-BET-762, or (-)-JQ1 (0.5 μM) for 48 h, followed by stimulation with lipopolysaccharide (LPS, from E. coli 055:B5, 1 μg/mL, for 24 h in the presence of the drugs).

## Messenger RNA determination

Total RNA was isolated from cells cultured in T25 flasks using the RNeasy Mini Kit and the RNase-Free DNase Set (Qiagen, Chadstone Centre, VIC Australia) following the manufacturer's protocols. RNA purity was assessed by UV absorption using a NanoDrop 1000 spectrophotometer, and the Agilent 2100 Bioanalyzer System was used to verify RNA integrity. Purified RNA was spiked with Alien RNA transcript ($10^7$ copies/μg, Stratagene, Integrated Sciences, Chatswood NSW Australia), which served as the reference RNA in the qRT-PCR assays.

The RNA was reverse transcribed using the Superscript™ III (Life Technologies, Mulgrave, VIC Australia) with random hexamer primers. Quantitative real-time PCR was performed using the Applied Biosystems™ QuantStudio™ 6-Flex Real-Time PCR system (Thermo Fisher, Scoresby VIC Australia) with SYBR™ Green detection reagents sourced from the manufacturer. Primers have been designed using Primer-BLAST from NCBI, and their sequences are listed in the S1 Table. Primer and template cDNA concentrations were optimized for each mRNA assay [27].

## Chromatin immunoprecipitation (ChIP)

**Cell fixation.** Cells grown in T75 culture flasks were fixed by a two-step cross-linking procedure that enhances the detection of modified histones and transcription factors bound to the chromatin [28]. Briefly, the adherent cultures were rinsed with PBS (phosphate-buffered saline) and cross-linked by adding 2 mM EGS (Ethylene glycol-bis(succinic acid N-hydroxysuccinimide ester) (Sigma-Aldrich E3257) in PBS for 30 min at room temperature. Next, fresh buffer with 1% formaldehyde was added at room temperature for 10 min to perform the second cross-linking step. Formaldehyde action was quenched with 125 mM glycine in cold PBS. Cells were permeabilized in hypotonic buffer (50 mM Tris-HCl, pH 7.5, 10 mM NaCl, 3 mM $MgCl_2$, 0.15% v/v Igepal Ca-630) supplemented with protease inhibitors (cOmplete™ Mini Protease Inhibitor Cocktail, Roche, and 0.5 mM phenylmethanesulfonyl fluoride, PMS) and histone deacetylase inhibitor (10 mM Na-butyrate). Cells were scraped off the culture surface, and the nuclei were sedimented by centrifugation and stored at -80˚C until further processing.

**Preparation of chromatin extracts.** Nuclear pellets were suspended in Extraction Buffer (50 mM Tris HCl, pH8.1; 10 mM EDTA; 1% SDS and inhibitors as above) and sonicated using a Misonix XL-2000 microprobe sonifier with the following optimized settings: Ten 10-sec bursts at 8–11 watts and one pulse/sec, separated by 50-sec intervals. Heating was prevented by performing the sonication in a salt-ice bath. Sonicated samples were centrifuged at 13,000 rpm at 4˚C for 30 min, and the supernatants were stored frozen until immunoprecipitation.

**Immunoprecipitation.** Chromatin extracts were diluted 10-fold with Modified RIPA buffer (1.13% Triton X-100, 0.11% Na-deoxycholate, 1.1 mM EDTA, 0.56 mM EGTA, 158

mM NaCl, 10mM Tris HCl pH 8.1 supplemented with 0.55 mM PMSF, 0.11% BSA, 4.4 μg/mL sonicated salmon sperm DNA, 11.1 mM Na-butyrate, cOmplete protease inhibitors). Immunoprecipitations were performed in 500 μL volume at 4˚C overnight using antibodies (3μg/ immunoprecipitation) listed in S2 Table. Immuno-complexes were captured by Protein-A/G Magnetic Beads (Pierce™, ChIP-grade, Cat. No. 26162) pre-washed and pre-equilibrated with Modified RIPA diluted with Extraction Buffer (9:1). Beads with attached immunocomplexes were collected in a magnetic stand and washed sequentially with the following solutions: Once with IPW (0.1% SDS, 1% Triton X100, 2 mM EDTA, 20 mM Tris HCl pH 8.1, 150 mM NaCl), once with IPW containing 500 mM NaCl, once with IPWL (0.25M LiCl, 1% NP40, 1% (w/v) Na-deoxycholate, 1mM EDTA, 10mM Tris HCl, pH 8.1) and twice with TE (10mM Tris, pH 8.1, 1 mM EDTA). Beads suspended in TE were transferred to clean Eppendorf LoBind tubes, reducing the background ChIP signal [29].

**Chromatin elution, crosslink reversal, and DNA purification.** Chromatin fragments were eluted by incubating the beads in 1% SDS, 0.1 M NaHCO$_3$ at 65˚C for 10 min. The bead-free eluates were supplemented with Proteinase K (57 μg/mL) and incubated at 56˚C for one hour, followed by 65˚C overnight to reverse crosslinks. An aliquot of each sonicated extract was processed for crosslink reversal without immunoprecipitation to determine the input in the ChIP assays.

DNA was purified by phenol-chloroform-isoamyl alcohol extraction (25:24:1 v/v) followed by Na-acetate-ethanol precipitation with 45 μg GlycoBlue™ (Invitrogen) co-precipitant added to each sample. Precipitates were washed 2 x with TE buffer to remove residual SDS and dissolved in TE. DNA content in the input samples was determined by UV absorption using the NanoDrop 1000 spectrophotometer.

**Quantitative real-time PCR (qPCR).** The abundance of target gene promoter sequences in the immunoprecipitated DNA was determined by qPCR. Gene-specific oligonucleotide primer sequences and their positions relative to transcription start sites are listed in S1 Table. Oligonucleotide primers were designed and optimized as described above. Triplicate PCR reactions were performed with each chromatin extract and gene from each immunoprecipitation using the QuantStudio™ 6-Flex Real-Time PCR system with SYBR™ Green detection.

## Data processing and statistical analysis

**Messenger RNA abundance.** Results were calculated relative to the spiked-in exogenous RNA [30] (Alien) by the delta-Ct method [27]. Biological variation was reduced by scaling data over the average of all treatments in each experiment. Data were tested for distribution and transformed to meet the criteria for normal distribution. The data were then analyzed by repeated-measures ANOVA followed by F-tests for individual contrasts with significance adjusted to false discovery rate (FDR, "Q") using the Benjamini-Hochberg method. In some of the experiments, as indicated in the Figure legends, the non-normally distributed data were evaluated using the non-parametric K-sample-equality-of-medians test followed by Fisher's exact test, Wilcoxon's signed-rank test, or Dunn's test of multiple comparisons using rank sums with Bonferroni-adjustment of significance, as appropriate. Differences at $p < 0.05$ or $Q < 0.05$ were considered statistically significant.

**ChIP.** Results were calculated as the recovery of promoter fragments relative to input using delta-Ct values. The results for each gene were scaled over the average of all treatments with each immunoprecipitation (acetyl-histone 3, acetyl-histone 4, BRD2, BRD4, IgG) in each experiment and expressed relative to the negative control, IgG, to reduce biological and technical variation. Data were tested for distribution and transformed to meet normal distribution criteria. The data were then analyzed by repeated-measures ANOVA followed by F-tests for

individual contrasts with significance adjusted to false discovery rate (FDR, "Q") using the Benjamini-Hochberg method. Q<0.05 (5% FDR) was considered statistically significant. Statistical calculations were performed using STATA/IC 15.1 (College Station, TX; RRID: SCR_012763)

## Results

### BETs are involved in gene expression control in DSCs

**Treatments and gene targets.** We have utilized two highly selective BET inhibitors, (+)-JQ1 and I-BET-762 [31, 32], which can serve as chemical probes of functional BET involvement in chromatin regulation and gene expression control [33]. We have also treated cells with (-)-JQ1, the enantiomer of (+)-JQ1, as the inactive negative control compound [31]. Lipopolysaccharide (LPS) was used to model inflammatory stimulation by bacteria. We determined the effects of the treatments on a panel of pro-and anti-inflammatory genes selected for their potential participation in the inflammatory response of DSCs (S3 Fig). Basal levels of these mRNAs varied significantly in vehicle-treated cells (p = 0.001, K-sample-equality-of-medians test), with the constitutively expressed *PTGS1* and *PTGES* exhibiting the highest mRNA abundance. At the same time, the expression of the inducible *PTGS2*, *TNF*, and *IDO1* genes was low.

**The effects of LPS and BET inhibitors on target gene expression.** Fig 1 summarizes the effects of LPS, BET inhibitors, and the combination of LPS and BET inhibitors on target gene expression in a heat map grid format. Significant stimulation and inhibition at 5% FDR are visualized in the orange and blue cells of the grid, respectively, while non-significant contrasts are shown in the uncolored cells. The FDR values were calculated using all contrasts (28 altogether) presented in the heat map in S4 Fig with the uncorrected P-values. Relative mRNA abundances for each gene and responses to treatments are displayed individually in Fig 2 (*PTGS1*, *PTGES*, and *PTGS2*, encoding members of the prostaglandin synthesis pathway), Fig 3 (*IL6*, *CXCL8/IL8*, and *TNF*, encoding key proinflammatory mediators), and Fig 4 (*IL10* and *IDO1*, encoding anti-inflammatory mediators). Significant differences highlighted in the heatmap in Fig 1 are indicated in the figures. LPS significantly increased the mRNA levels (*vs.* vehicle) of the genes selected for inclusion in the panel, except for *PTGS1* and *PTGES*. The BET

| Contrasts | Q (5% FDR) | | | | | | | |
|---|---|---|---|---|---|---|---|---|
| | PTGS1 | PTGES | PTGS2 | IL6 | CXCL8/IL8 | TNF | IL10 | IDO1 |
| LPS vs. Veh | 0.2780 | 0.1073 | **<0.0001** | **<0.0001** | **<0.0001** | **<0.0001** | **0.0448** | **<0.0001** |
| (+)JQ1 vs. Veh | **0.0385** | 0.7238 | **<0.0001** | **<0.0001** | 0.3421 | 0.7550 | **0.0001** | 0.0533 |
| (-)JQ1 vs. Veh | 0.9940 | 0.8642 | 0.9810 | 0.9810 | 0.9830 | 0.3250 | 0.3187 | 0.2057 |
| I-BET-762 vs. Veh | 0.2387 | 0.9150 | **<0.0001** | **<0.0001** | 0.2067 | 0.1792 | **0.0003** | 0.2722 |
| LPS+(+)JQ1 vs. LPS | **0.0020** | 0.3696 | **<0.0001** | **<0.0001** | **0.0406** | 0.1606 | **<0.0001** | **0.0056** |
| LPS+(-)JQ1 vs. LPS | 0.2619 | 0.8950 | 0.9790 | 0.9790 | 0.9281 | 0.2453 | 0.9750 | 0.7580 |
| LPS+I-BET-762 vs. LPS | 0.0700 | 0.3640 | **0.0006** | **0.0006** | **0.0311** | 0.6668 | **<0.0001** | **0.0378** |

| KEY | |
|---|---|
| | Increase |
| | Decrease |

**Fig 1. Heat map grid summarizing the effects of LPS and BET inhibitors on target gene mRNA expression in DSCs.** Culture and treatment conditions, qRT-PCR assay details, and data processing are described in the Materials and methods section. The grid cells show significance values (Q) adjusted to the false discovery rate (FDR). The significance threshold is 5% FDR (Q<0.05), highlighted in bold. The orange color indicates a significant increase, the blue color indicates a significant decrease regarding the contrasts shown. The gene symbols are in red. The Q values have been calculated by contrasting all treatments with each other (28 contrasts, listed in S4 Fig). The contrasts selected for display in the heatmap indicate LPS effects (LPS *vs.* Vehicle), drug effects ((+)-JQ1 or (-)-JQ1 or I-BET-762 *vs.* Vehicle), and drug effects on the LPS response (LPS + (+)-JQ1 or (-)-JQ1 or I-BET-762 *vs.* LPS).

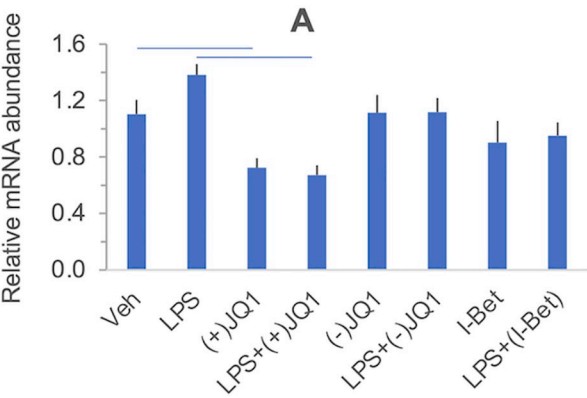

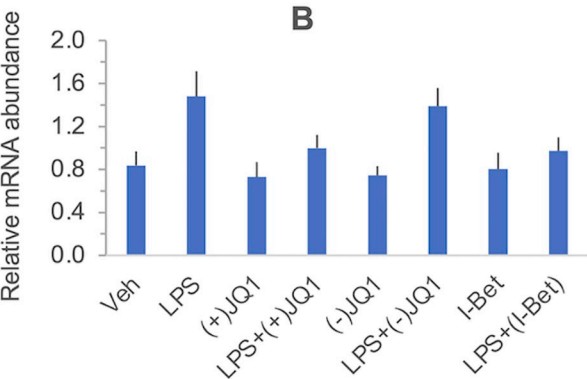

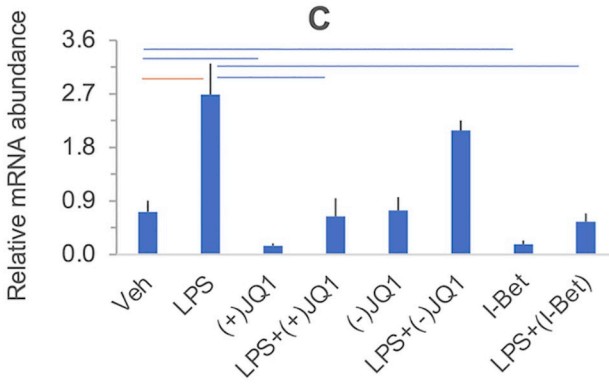

**Fig 2.** The effect of LPS and BET inhibitors on *PTGS1* (Panel A), *PTGES* (Panel B) and *PTGS2* (Panel C) mRNA expression in DSCs. Sub-confluent primary DSC cultures were treated with vehicle, LPS, (+)-JQ1, (-)-JQ1, I-BET-762, and their combinations as indicated under the columns. Messenger RNA abundance was determined by qRT-PCR. Further details of the experimental procedures and data analysis are described in the Materials and methods. Columns show the mean ± SEM, n = 4–5 independent experiments. Significant differences (at 5% FDR), highlighted in the heatmap in Fig 1, are indicated by the horizontal orange and blue lines. The results of the individual experiments and summary statistics are presented in the S1 Data.

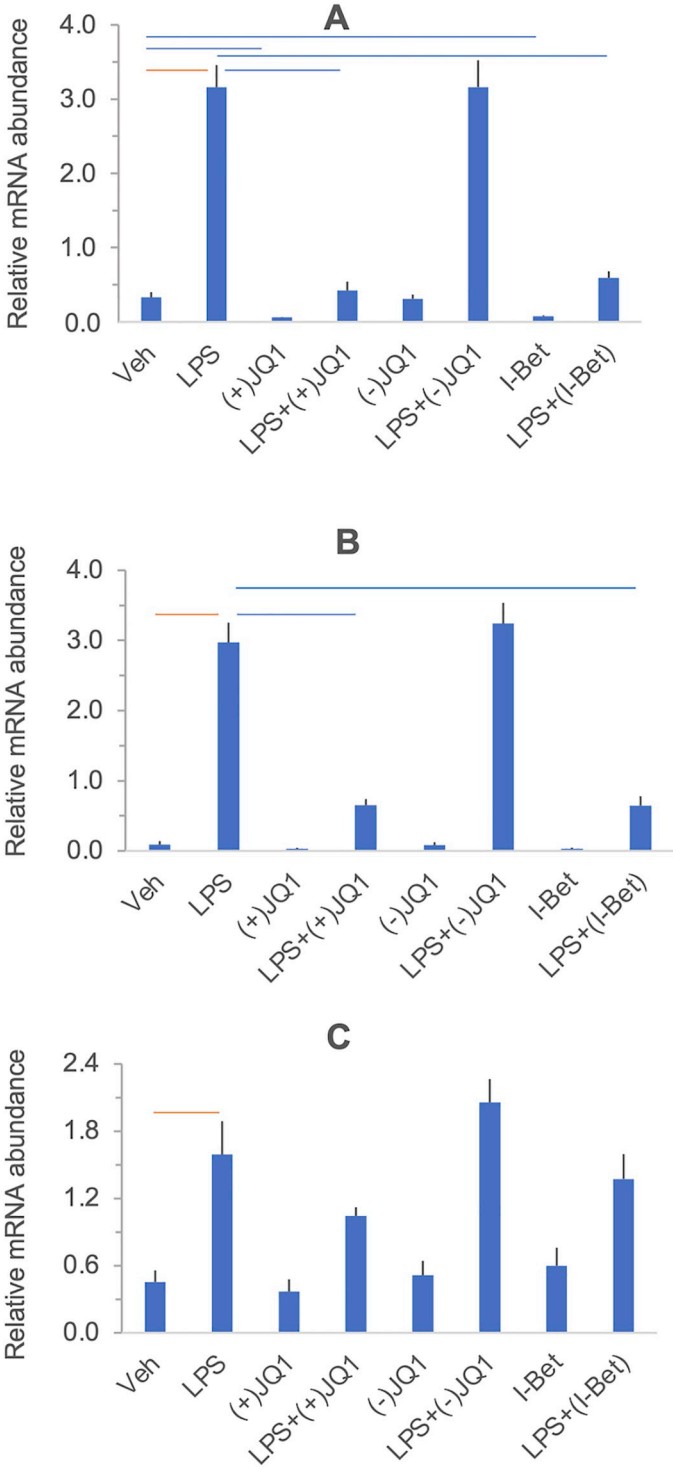

**Fig 3.** The effect of LPS and BET inhibitors on *IL6* (Panel A), *CXCL8/IL8* (Panel B) and *TNF* (Panel C) mRNA expression in DSCs. Primary DSC cultures were treated with vehicle, LPS, (+)-JQ1, (-)-JQ1, I-BET-762, and their combinations as indicated under the columns. Messenger RNA relative abundance was determined by qRT-PCR. Significant differences (at 5% FDR), highlighted in the heatmap in Fig 1, are indicated by the horizontal orange and blue lines. Further details and data presentation are described in Legend to Fig 2.

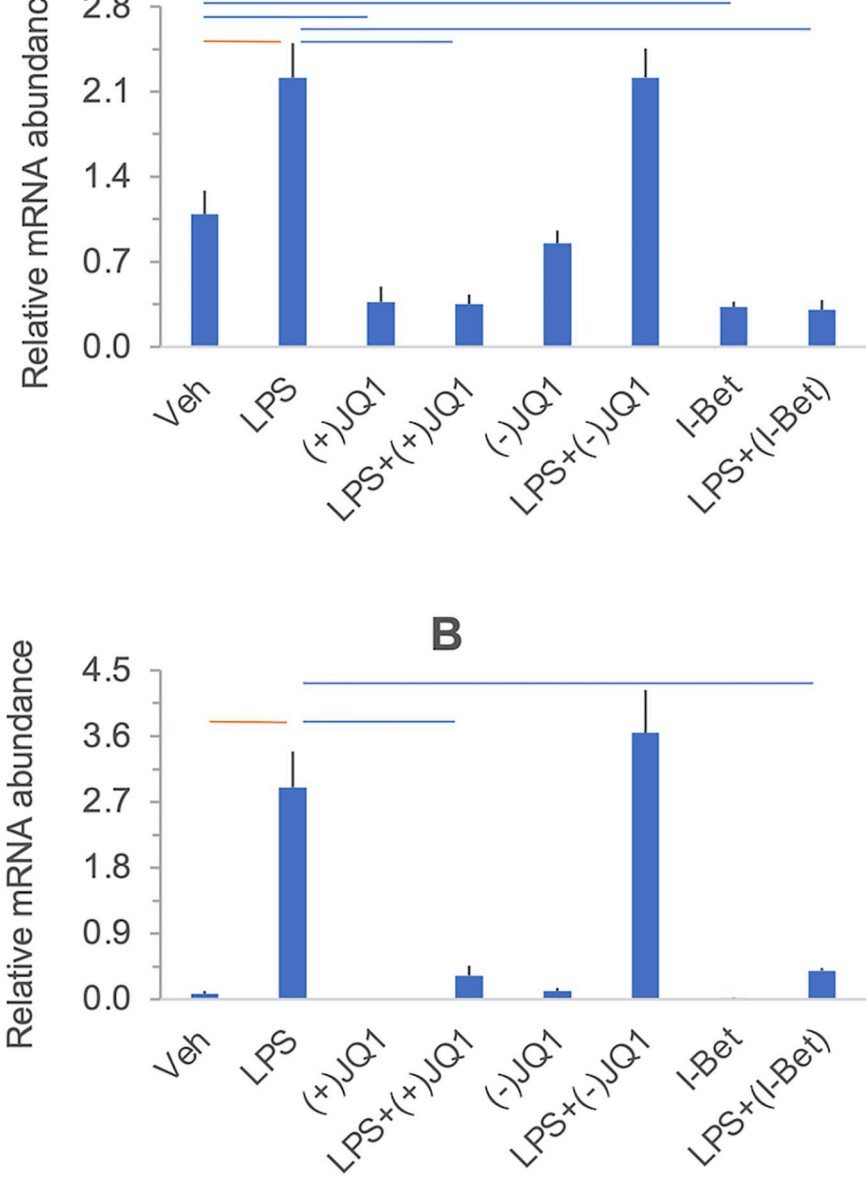

**Fig 4.** The effect of LPS and BET inhibitors on *IL10* (Panel A) and *IDO1* (Panel B) mRNA expression in DSCs. Primary DSC cultures were treated with vehicle, LPS, (+)-JQ1, (-)-JQ1, I-BET-762, and their combinations as indicated under the columns. Messenger RNA relative abundance was determined by qRT-PCR. Significant differences (at 5% FDR), highlighted in the heatmap in Fig 1, are indicated by the horizontal orange and blue lines. Further details and data presentation are described in Legend to Fig 2.

inhibitors (+)-JQ1 and I-BET-762, but not the negative control (-)-JQ1, reduced the basal (drugs vs. vehicle) and LPS-stimulated (LPS + drugs *vs.* LPS) expression of *PTGS2*, *IL6*, and *IL10*. The drugs did not affect the low basal levels of *CXCL8/IL8* and *IDO1* mRNAs. Still, the LPS-induced expression of these genes was significantly blocked by the BET inhibitors but not the control compound. BET inhibitors did not affect the expression of *TNF* and *PTGES* either in the presence or absence of LPS. Finally, the abundance of *PTGS1* mRNA was decreased by (+)-JQ1 both in the presence and absence of LPS, while (-)-JQ1 and I-BET-762 had no effect

under the treatment conditions employed. The response patterns of *PTGS2*, *IL6*, *IL10*, *CXCL8/IL8*, and *IDO1* to BET inhibitors strongly suggest that BET proteins are involved in the mechanism of LPS stimulation of these genes in the DSCs. The lack of effect of BET inhibitors indicates that *TNF* and *PTGES* expression is not BET dependent. Our data suggest some BET involvement in the (constitutive) expression of PTGS1.

## BETs are expressed in DSCs

(+)-JQ1 and I-BET-762 are pharmacological probes targeting the tandem bromodomains of the BET family members BRD2, BRD3, BRD4, and BRDT (bromodomain-containing proteins 2, -3, -4, and bromodomain testis-specific protein) [33, 34]. We have determined the expression of these BETs (except for the testis-specific BRDT) in DSCs to delineate the BET inhibitor targets that mediate the effects of the drugs. Data in Fig 5A) shows that *BRD2*, *BRD3*, and *BRD4* mRNAs are present in DSCs. Both the long (canonical) *BRD4* mRNA variant, *BRDL*, and the truncated *BRD4S* were detected. Differences in the mRNA levels indicated that BRD2 and BRD4L were the dominant BETs in the DSCs. LPS did not affect *BRD2* and *BRD4L* expression (Fig 5B and 5C). Treatment with (+)-JQ1, but not with (-)-JQ1, significantly increased the level of *BRD2* mRNA (Fig 5B), indicating BET-dependent negative feedback. *BRD4L* mRNA levels were not affected by the drugs (Fig 5C).

## Histone acetylation and BET presence at target gene promoters

BET proteins bind acetylated histone 3 and -4 (acH3 and acH4) through their bromodomains, which is critical to their actions in chromatin. BRD2 and -4 have been reported to bind and interact with the promoters of genes they control [35, 36]. Based on these observations, we have explored histone acetylation and BRD2 and -4 presence at the promoters of the genes induced by LPS in our gene panel. Chromatin immunoprecipitation (ChIP) was performed using antibodies that recognize acH3, acH4, BRD2, and BRD4, and the enrichment of proximal promoter sequences of the target genes relative to the negative control IgG was determined in the immunoprecipitated chromatin fragments. Enrichment levels were compared among cultures treated with vehicle, LPS, (+)-JQ1, (-)-JQ1, and their combinations.

**Acetyl histone 3.**   The heat map grid in Fig 6 summarizes the changes of acH3 levels at the gene promoters in cells treated with LPS, (+)-JQ1 and (-)-JQ1. Significance thresholds were adjusted to 5% FDR (Q) based on all contrasts, shown in S5 Fig, including the unadjusted P values. Figs 7 and 8 display the acH3 ChIP results in each individual gene promoter under each treatment condition and indicate the significant differences highlighted in the heatmap in Fig 6.

We have detected histone 3 acetylation at the *IL6*, *CXCL8/IL8*, and *TNF* promoters in unstimulated DSCs (vehicle *vs*. the ChIP negative control, IgG). Histone 3 acetylation was not detected at the promoters of the anti-inflammatory *IL10* and *IDO1* genes in unstimulated cells. LPS treatment induced histone 3 acetylation at the *IDO1* promoter but not at the other promoters (LPS *vs*. Vehicle). Treatment with (+)-JQ1 alone resulted in a significant decrease in acH3 level at the *IL6* promoter, which was also seen in the presence of the negative control compound, (-)-JQ1 ((+)-JQ1 or (-)-JQ1 *vs*. Vehicle). Conversely, (+)-JQ1 alone increased histone 3 acetylation at the *IL10* and *IDO1* promoters, but this effect was not shared by (-)-JQ1. Treatment with (-)-JQ1, but not with (+)-JQ1, reduced acH3 levels at the *IL6* and *CXCL8/IL8* promoters after LPS exposure (LPS + (+)-JQ1 or (-)JQ1 *vs*. LPS). Furthermore, (+)-JQ1 increased histone 3 acetylation at the *TNF* and *IL10* promoters in response to LPS, but the negative control drug (-)-JQ1 was ineffective at these sites. At the *IDO1* promoter, histone 3 acetylation after LPS treatment was unaffected by the drugs. Overall, the differences in Histone 3

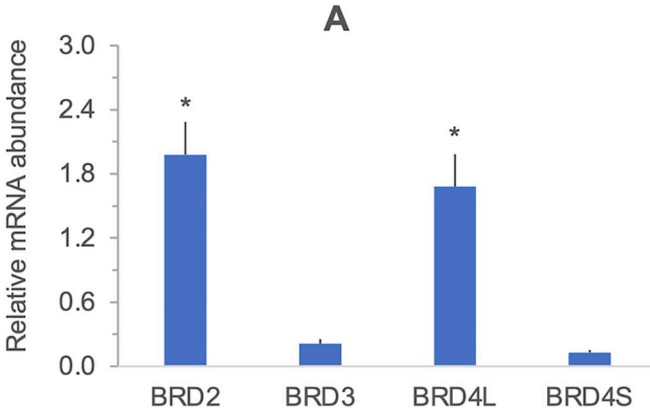

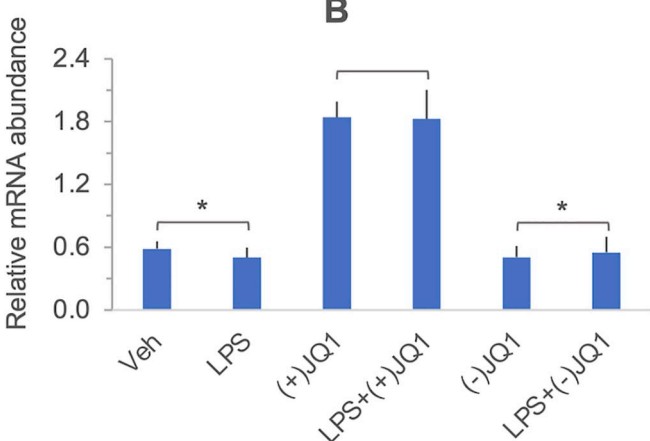

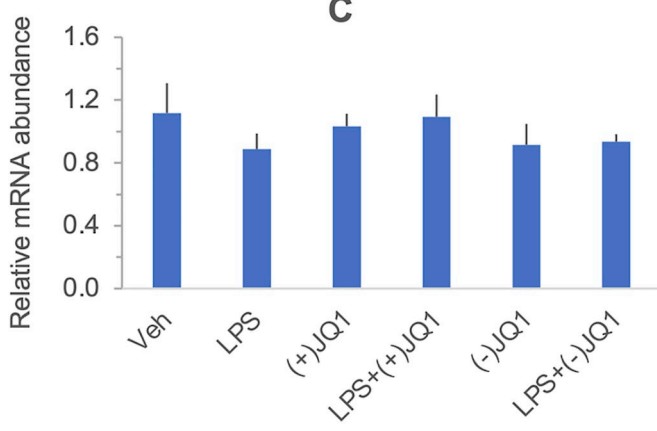

**Fig 5. Expression of BET family members in DSCs.** Total RNA was extracted from primary DSC cultures and mRNA levels encoding BRD2, BRD3, BRD4S and BRD4L were determined by qRT-PCR. (Panel **A**) Columns represent the mean ± SEM of n = 6 independent experiments. *, p<0.05 *vs.* BRD3 and BRD4S, by Dunn's test (non-parametric) with Bonferroni correction for multiple comparisons. Panels **B** and **C**: The effect of (+)-JQ1, (-)-JQ1, and LPS on *BRD2* and *BRD4L* expression, respectively. Cultures were treated with drugs and LPS, as described in the Materials and methods. Messenger RNA levels were determined by qRT-PCR. Columns represent the mean ± SEM of n = 4–9 independent experiments. LPS did not affect *BRD* mRNA levels (p>0.05, Wilcoxon's signed-rank test), and drug effects were evaluated after pooling data with LPS- treated and LPS-untreated cells. *, p<0.05 *vs.* (+)-JQ1 treated

cells (Dunn's test of multiple comparisons using rank sums with Bonferroni adjustment of significance). The results of the individual experiments and summary statistics are presented in the S1 Data.

acetylation showed no consistent relationship with the gene expression changes detected in response to the same treatments.

**Acetyl histone 4.** The histone 4 acetylation responses to LPS, (+)-JQ1, and (-)-JQ1 are summarized in the heatmap in Fig 9. As before, a heat map containing the full list of contrasts is presented in the S6 Fig. Acetyl-H4 ChIP results in each gene promoter under all treatment conditions are displayed in Figs 10 and 11, and significant differences highlighted in Fig 9 are indicated. In unstimulated cells, we have detected histone 4 acetylation at the *IL6* promoter (relative to IgG). After LPS stimulation, the acH4 levels increased significantly at the *TNF* and the *IDO1* promoters relative to the vehicle. Notably, the acH4 level at the *CXCL8/IL8* promoter was significant in LPS-treated cells relative to the IgG negative control but not compared to vehicle-treated cells, which had an acH4 level similar to IgG, suggesting indirectly that LPS stimulated histone 4 acetylation at this locus, as well. Treatment with (+)-JQ1 alone increased histone 4 acetylation of the *IL10* and the *IDO1* promoters relative to the vehicle, while (-)-JQ1 did not show this effect. At the IL6 promoter, (-)-JQ1, but not its active analog, (+)-JQ1, reduced histone 4 acetylation relative to vehicle treatment (Fig 10A). The drugs did not influence histone 4 acetylation at the *CXCL8/IL8*, *IL10*, and *IDO1* promoters in response to LPS. At the IL6 promoter, treatment with (-)-JQ1, but not with (+)-JQ1, reduced acH4 levels in response to LPS. Like with histone 3, histone 4 acetylation changes at the promoters showed no consistent relationship to the gene expression changes after the same treatments.

**BRD2 and BRD4.** ChIP analysis with antibodies verified for chromatin immunoprecipitation did not detect significant BRD2 or BRD4 binding to the promoter of the investigated genes at the 5% FDR level under our experimental conditions. The contrast lists of BRD2 and BRD4 ChIP results are provided in S7 and S10 Figs, respectively. The diagrams showing the

| Contrasts | Q (5% FDR) | | | | |
|---|---|---|---|---|---|
| | IL6 | CXCL8/IL8 | TNF | IL10 | IDO1 |
| Veh vs. IgG | **<0.0001** | **0.0027** | **0.0163** | 0.1950 | 0.9123 |
| LPS vs. Veh | 0.7750 | 0.1414 | 0.1875 | 0.6857 | **0.0004** |
| LPS vs. IgG | **<0.0001** | **<0.0001** | **0.0005** | 0.3636 | **0.0010** |
| (+)JQ1 vs. Veh | **0.0251** | 0.1719 | 0.2602 | **0.0303** | **0.0010** |
| (-)JQ1 vs. Veh | **0.0004** | 0.1140 | 0.2597 | 0.6886 | 0.1549 |
| LPS + (+)JQ1 vs. LPS | 0.5372 | 0.2557 | **0.0060** | **0.0024** | 0.1638 |
| LPS + (-)JQ1 vs. LPS | **<0.0001** | **0.0053** | 0.2186 | 0.0893 | 0.9770 |

| KEY | |
|---|---|
| | Increase |
| | Decrease |

**Fig 6. Heat map grid summarizing the effects of LPS, (+)-JQ and (-)-JQ1 on histone 3 acetylation at the promoters of *IL6*, *CXCL8/IL8*, *TNF*, *IL10* and *IDO1*.** Culture and treatment conditions, chromatin immunoprecipitation (ChIP) details, and data processing are described in the Materials and methods section. The grid cells show significance values (Q) adjusted to the false discovery rate (FDR). The significance threshold is 5% FDR (Q<0.05), highlighted in bold. The orange color indicates a significant increase, the blue color indicates a significant decrease in the acH3 ChIP signal regarding the contrasts shown. The gene symbols are in red. The Q values have been calculated by contrasting all treatments with each other (21 contrasts, listed in S5 Fig). The contrasts selected for display in the heatmap indicate LPS effects (LPS *vs.* Vehicle), drug effects ((+)-JQ1 or (-)-JQ1 *vs.* Vehicle), and drug effects on the LPS response (LPS + (+)-JQ1 or (-)-JQ1 *vs.* LPS).

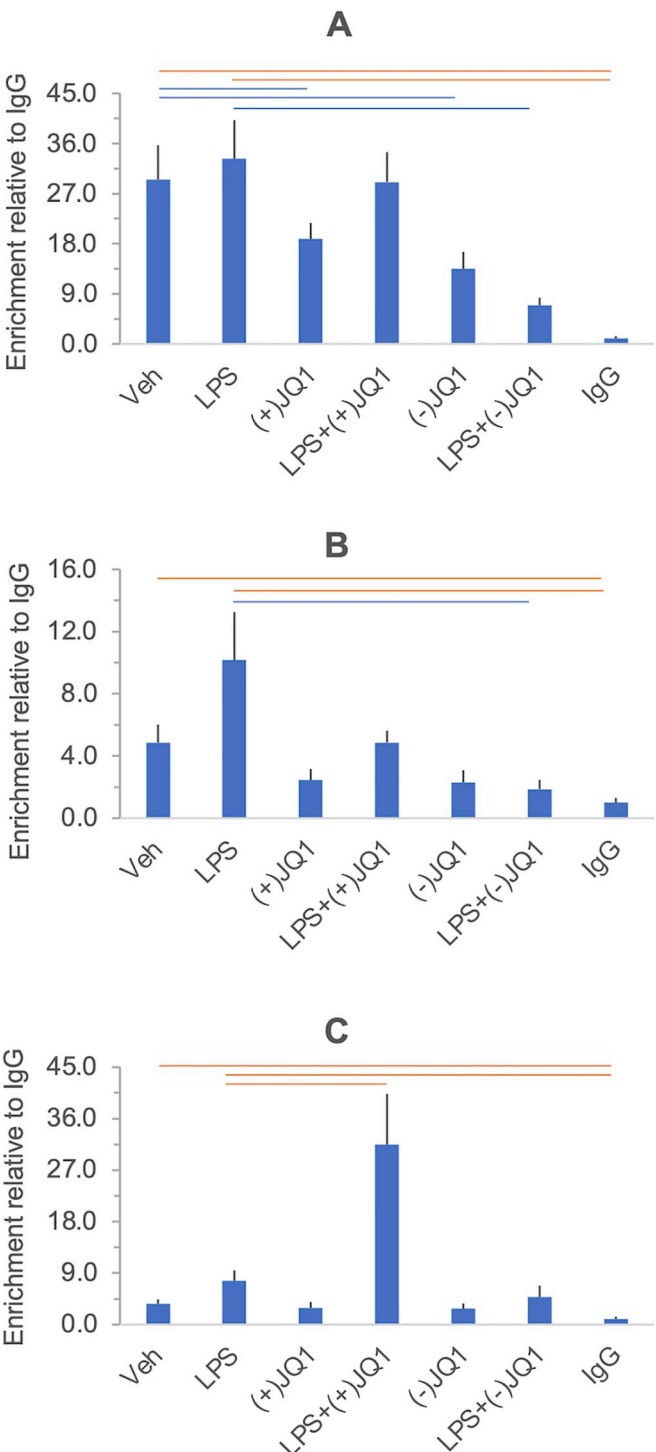

**Fig 7.** The effects of LPS, (+)-JQ1 and (-)-JQ1 on histone 3 acetylation at the *IL6* (Panel A) *CXCL8/IL8* (Panel B) and *TNF* (Panel C) promoters in DSCs. Primary cultures of DSCs were treated with vehicle, LPS, (+)-JQ1, (-)-JQ1, and their combinations indicated under the columns. Histone 3 acetylation was determined by chromatin immunoprecipitation (ChIP) using a pan-acetyl histone 3 antibody (listed in S2 Table), and the results were expressed relative to the negative control IgG. Further details of the experimental procedures and the data processing are described in the Materials and methods section. Columns show the mean ± SEM, n = 5–6 independent experiments. Significant differences (at 5% FDR), highlighted in the heatmap in Fig 6, are indicated by the horizontal orange and blue lines. The results of the individual experiments and summary statistics are presented in the S1 Data.

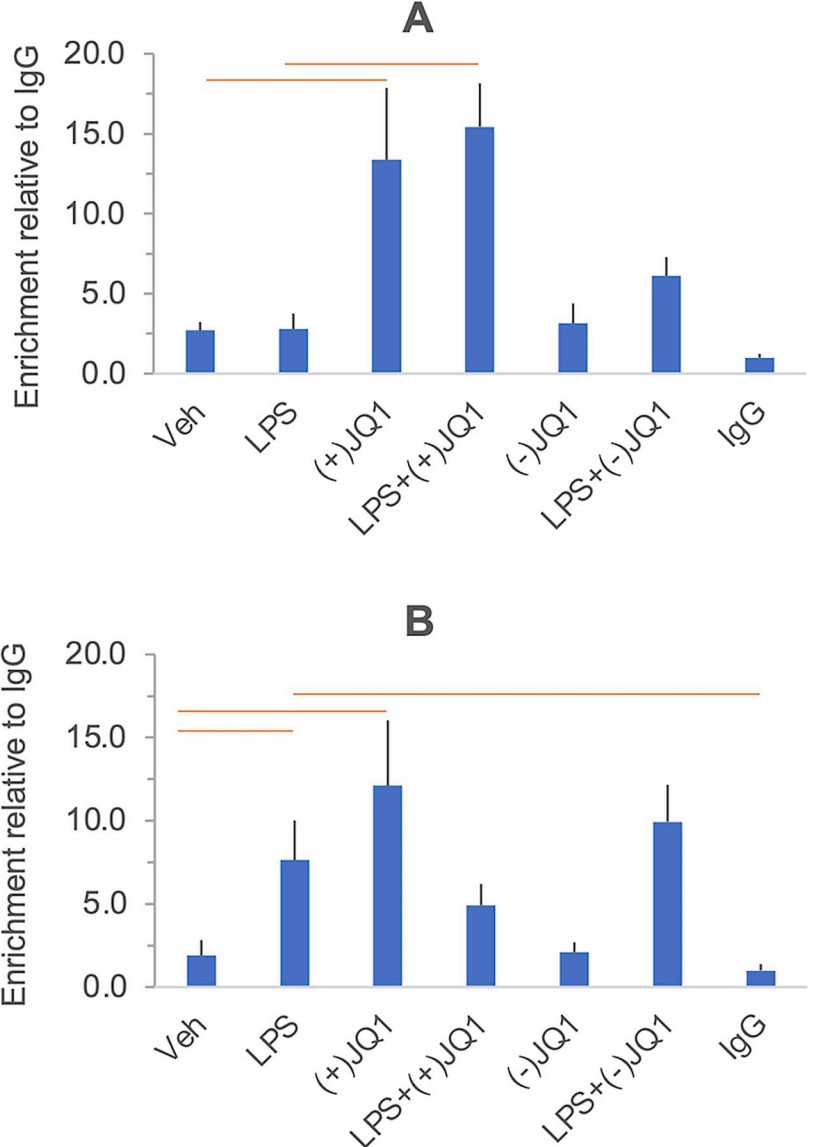

**Fig 8.** The effects of LPS, (+)-JQ1, and (-)-JQ1 on histone 3 acetylation at the *IL10* (Panel A) and *IDO1* (Panel B) promoters in DSCs. Primary cultures of DSCs were treated with vehicle, LPS, (+)-JQ1, (-)-JQ1, and their combinations as indicated under the columns. Histone 3 acetylation was determined by chromatin immunoprecipitation (ChIP). Columns show the mean ± SEM, n = 5–6 independent experiments. Significant differences (at 5% FDR), highlighted in the heatmap in Fig 6, are indicated by the horizontal orange and blue lines. Further details are described in the legend in Fig 7. The results of the individual experiments and summary statistics are presented in the S1 Data.

BRD2 and BRD4 ChIP results in individual gene promoters under all treatment conditions are presented in S8 and S9 Figs for BRD2 and S11 and S12 Figs for BRD4.

## Discussion

Pharmacological agents that bind the tandem bromodomains of the BET family transcription factors have been used successfully to determine the involvement of these proteins in regulatory events at the molecular level. Blocking the bromodomains interferes with the binding of BETs to lysine-acetylated histones disrupting their gene regulatory function. Two of the best-

| Contrasts | Q (5% FDR) | | | | |
|---|---|---|---|---|---|
| | IL6 | CXCL8/IL8 | TNF | IL10 | IDO1 |
| Veh vs. IgG | **<0.0001** | 0.3265 | 0.6980 | 0.9630 | 0.7581 |
| LPS vs. Veh | 0.7025 | 0.0714 | **0.0210** | 0.1661 | **0.0244** |
| LPS vs. IgG | **<0.0001** | **0.0120** | **0.0126** | 0.1663 | **0.0105** |
| (+)JQ1 vs. Veh | 0.4858 | 0.0665 | 0.0821 | **0.0350** | **0.0075** |
| (-)JQ1 vs. Veh | **0.0113** | 0.6377 | 0.6668 | 0.7721 | 0.7781 |
| LPS + (+)JQ1 vs. LPS | 0.4594 | 0.5329 | 0.0525 | 0.1785 | 0.4256 |
| LPS + (-)JQ1 vs. LPS | **<0.0001** | 0.3554 | 0.1965 | 0.5236 | 0.5787 |

| KEY | |
|---|---|
| | Increase |
| | Decrease |

**Fig 9. Heat map grid summarizing the effects of LPS, (+)-JQ, and (-)-JQ1 on histone 4 acetylation at the promoters of *IL6, CXCL8/IL8, TNF, IL10*, and *IDO1*.** Culture and treatment conditions, chromatin immunoprecipitation (ChIP) details, and data processing are described in the Materials and methods section. The grid cells show significance values (Q) adjusted to the false discovery rate (FDR). The significance threshold is 5% FDR (Q<0.05), highlighted in bold. The orange color indicates a significant increase, the blue color indicates a significant decrease in the acH3 ChIP signal regarding the contrasts shown. The gene symbols are in red. The Q values have been calculated by contrasting all treatments (21 contrasts, listed in S6 Fig). The contrasts selected for display in the heatmap indicate LPS effects (LPS *vs.* Vehicle), drug effects ((+)-JQ1 or (-)-JQ1 *vs.* Vehicle), and drug effects on the LPS response (LPS + (+)-JQ1 or LPS + (-)-JQ1 *vs.* LPS).

characterized members of the BET inhibitors are (+)-JQ1 and I-BET-762. When these compounds are employed at a concentration of $\leq 1$ µM, and their effects are contrasted to an analog of a similar structure practically inactive on the target, conclusions about functional BET involvement can be drawn with confidence. The verified control compound we used was (-)-JQ1, the enantiomer of (+)-JQ1 [31, 37]. The pharmacological selectivity of BET inhibitors provided an opportunity to assess the involvement of the BET proteins in the regulation of inflammatory genes in DSCs critical for pregnancy maintenance and the timing of labor [7, 38, 39].

We have established primary cultures of decidual stromal cells isolated from term placentas by adapting published procedures [10]. Cells were purified by removing CD45[+] leukocytes and cultured without passage in the presence of estrogen and progestogen for a maximum of 20 days to approach conditions preserving the *in vivo* DSC phenotype. LPS was employed as the inflammatory stimulus to model infectious insults the decidua may be exposed to under pathological conditions. We chose to determine treatment effects on genes prototypical of pro-and anti-inflammatory responses. The target genes included *PTGS1, PTGS2*, and *PTGES*, coding for Prostaglandin G/H Synthase 1, -2; and Prostaglandin E Synthase, which are critical enzymes of the prostaglandin synthesis pathway; *IL6* (encoding Interleukin-6, a multifunctional proinflammatory cytokine), *CXCL8/IL8* (responsible for producing Interleukin-8, a neutrophil chemokine and proinflammatory factor), and *TNF* (the gene for Tumor Necrosis Factor, a pleiotropic cytokine critically involved in inflammatory responses). We have also studied genes encoding anti-inflammatory factors such as IL-10 (Interleukin-10) and IDO-1 (Indoleamine 2,3-Dioxygenase 1), a secreted tryptophan metabolizing enzyme with potent immune-mitigating and antimicrobial actions [40].

LPS stimulated the expression of all studied genes except *PTGS1* and *PTGES*, which had constitutively high mRNA levels. Notably, both the pro-inflammatory genes (*PTGS2, IL6, CXCL8/IL8, TNF*) and the anti-inflammatory genes (*IL10 and IDO1*) in our panel responded to LPS treatment. The responses were robust, even considering that the primary cells could

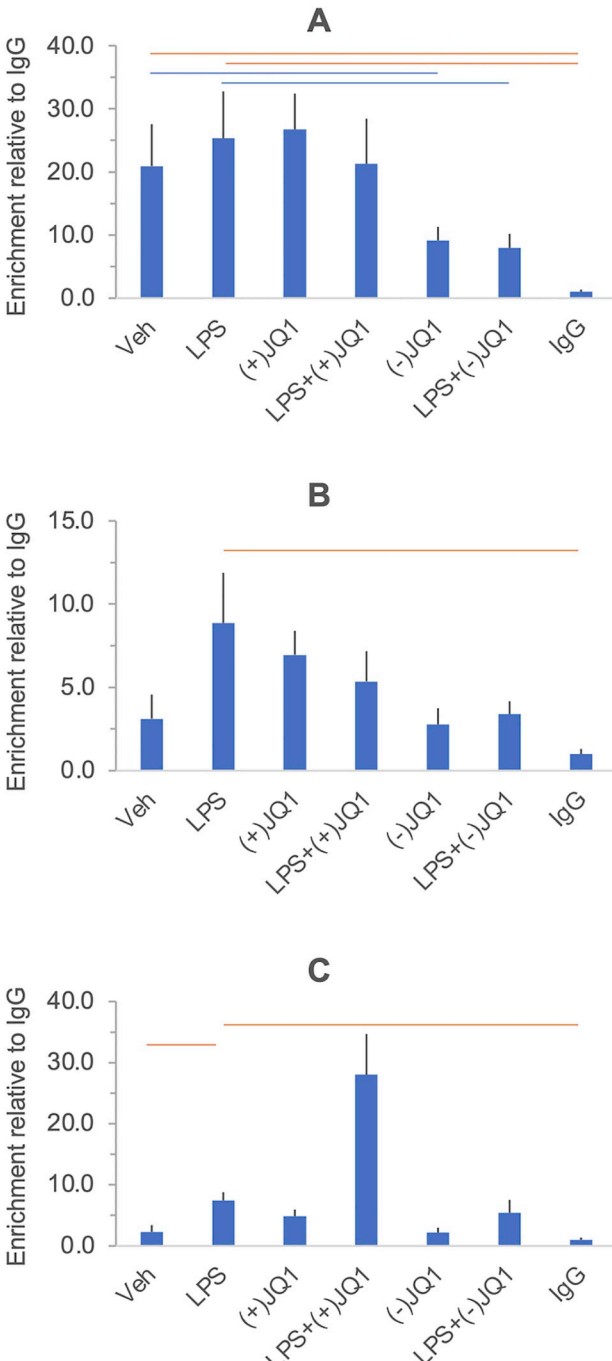

**Fig 10.** The effects of LPS, (+)-JQ1 and (-)-JQ1 on histone 4 acetylation at the *IL6* (Panel A) *CXCL8/IL8* (Panel B) and *TNF* (Panel C) promoters in DSCs. Primary cultures of DSCs were treated with vehicle, LPS, (+)-JQ1, (-)-JQ1, and their combinations indicated under the columns. Chromatin immunoprecipitation (ChIP) was performed using a pan-acetyl histone 4 antibody (S2 Table), and the results were expressed relative to the negative control IgG. Further details of the experimental procedures and the data processing are described in the legend of Fig 7. Columns show the mean ± SEM, n = 5–6 independent experiments. Significant differences (at 5% FDR), highlighted in the heatmap in Fig 9, are indicated by the horizontal orange and blue lines. The results of the individual experiments and summary statistics are presented in the S1 Data.

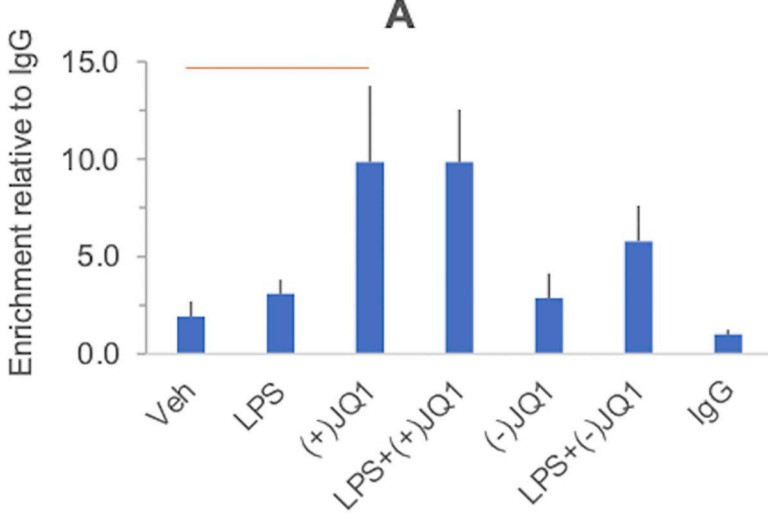

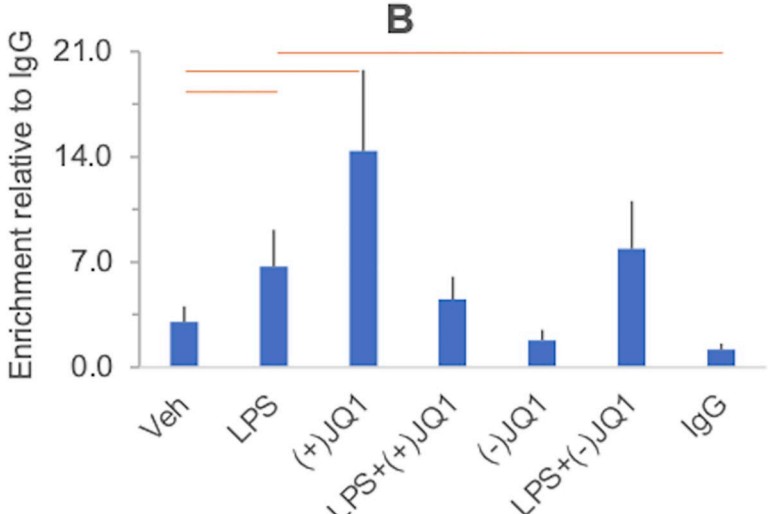

**Fig 11.** The effects of LPS, (+)-JQ1 and (-)-JQ1 on histone 4 acetylation at the *IL10* (Panel A) and *IDO1* (Panel B) promoters in DSCs. Experimental details and data presentation are the same as in Fig 10. Columns show the mean ± SEM, n = 5–6 independent experiments. Significant differences (at 5% FDR), highlighted in the heatmap in Fig 9, are indicated by the horizontal orange and blue lines. The results of the individual experiments and summary statistics are presented in the S1 Data.

have been exposed to infectious stimuli *in situ* before isolation. We took care to process tissues after elective operative (Cesarean) deliveries without clinical or histological signs of intrauterine infection or inflammation to minimize the risk of *in situ* LPS exposure. Nevertheless, this possibility represents a limitation of the study. The BET inhibitor effects on mRNA levels were consistent with the involvement of BETs in the LPS responses except for *TNF*, which was not influenced significantly by (+)-JQ1, I-BET-762, or (-)-JQ1. *PTGS2*, *IL6*, and *IL10* mRNA levels were reduced by (+)-JQ1 and I-BET-734, but not (-)-JQ1, in unstimulated cells, indicating BET involvement in basal expression. The lack of BET inhibitor effect on *TNF* expression suggests that there is a BET-independent pathway of LPS action in DSCs. TNF-a has been

reported to stimulate the IL-6 [41], IL-8 [9], IL-10 [42], and prostaglandin [42–45] production of decidual cells or tissue *in vitro*, and time course and immunoneutralization experiments suggested that TNF-a may mediate or augment the action of LPS on inflammatory mediator release from choriodecidua explants [42]. Moreover, TNF-a has been shown to promote the proinflammatory transformation of endothelial (HUVEC) cells by a mechanism that includes the redistribution of BRD4, one of the BETs dominantly expressed in decidual cells, to super-enhancer chromatin loci controlling inflammatory genes [19]. Our data are compatible with an LPS-induced, TNF-a-mediated, and BRD4 -dependent mechanism of inflammatory transition of DSCs. Firm evidence for this scenario will require integrated genome-wide data of transcriptome changes, chromatin remodeling, and BET localization in response to LPS and other labor-associated stimulants.

The chromatin immunoprecipitation study aimed to gain molecular-level information about BET involvement in LPS actions on our inflammatory gene panel. BET involvement in gene regulation is diverse and multifaceted [22]. We have focused on the proximal promoter regions (within 1kb upstream of the transcription start site) where BRD2 and BRD4 were reported to control many genes [36, 46]. Nonetheless, we detected no significant binding of BRD2 or BRD4 at 5% FDR to the proximal promoter of the investigated genes. However, a low level of binding cannot be excluded because some unadjusted significance values were below the P = 0.05 cut-off (S5 and S7 Figs). The BRD2 and the BRD4 antibodies have been ChIP certified, and we have employed a dual fixation procedure to detect both direct and indirect (protein-to-protein mediated) binding to DNA [47]. We cannot exclude that technical reasons, e.g., epitope masking, may have contributed to the low ChIP signal. Furthermore, we have assayed BET binding after extended treatments optimized for mRNA changes, which could not detect possible early transient BRD binding event(s). Establishing the time course of BET binding to a broader array of candidate genes and regulatory chromatin regions after stimulation, especially enhancers and super-enhancers [21, 22], is a critical next approach to delineating the relationship between chromatin-bound BRDs and target gene expression in DSCs.

Histone 3 acetylation was detected at the promoters of three of the five genes examined (*IL6*, *CXCL8/IL8*, *TNF)*. LPS did not affect acH3 levels at these promoters or at the promoter of *IL10*, which was not acetylated in the unstimulated state. The expression of these genes, however, was significantly enhanced by LPS. In our limited panel of LPS target genes, only *IDO1* exhibited a concomitant increase in mRNA expression and promoter histone 3 acetylation after endotoxin treatment. This indicates that histone 3 acetylation at the promoter is not a general obligatory requirement for LPS-induced inflammatory gene expression in DSCs. A similar conclusion can be drawn about histone 4 acetylation, which was found only at the *IL6* promoter in the unstimulated state and was induced by LPS only at the *TNF* and *CXCL8/IL8* promoters (vs. the IgG control) despite upregulated expression across the panel of genes analyzed by ChIP.

BET inhibitor treatment abrogated histone acetylation at the inflammatory gene panel. One type of response was observed at the *IL10* and *IDO1* promoters, where (+)-JQ1 increased H3 and H4 acetylation in unstimulated cells. Treatment with (+)-JQ1 enhanced H3 acetylation at the *IL10* and *TNF* promoters in response to LPS. (-)-JQ1 did not show these effects, arguing for BET involvement, which was likely indirect since no BRD binding was detected at these sites. BRD4 has been reported to possess an intrinsic histone acetyltransferase activity [48] and augment histone acetylation by interacting with p300 and CBP [49], but the mechanisms involved in DSCs remain to be established. Notably, acetyl-H3 and -H4 levels at the promoters showed no consistent relationship with gene expression changes. The second type of response was detected at the *IL6*, and *CXCL8/IL8* promoters, where the negative control compound, (-)-JQ1, but not its active analog, (+)-JQ1, reduced histone acetylation in LPS stimulated cells.

Furthermore, both (+)-JQ1 and (-)-JQ1 decreased *IL6* promoter acH3 levels in unstimulated cells. These actions of (-)-JQ1, even when copied by (+)-JQ1, may be off-target, not involving BETs. Alternatively, molecular dynamics (MD) simulations assessing the binding energies of (+)-JQ1 and (-)-JQ1 to the BRD4 bromodomain have indicated that both compounds possess similar affinities to BRD4 [50]. The more than 100-fold higher $IC_{50}$ of (-)-JQ1 compared to its chiral isomer, (+)-JQ1 [31], could be attributed to the faster insertion of (+)-JQ1 into the binding pocket of the bromodomain [50]. Despite the slower binding kinetics (-)-JQ1 might still be able to associate with BRDs after more prolonged exposure, such as 72 h in our experiments (48 h pre-treatment and 24 h co-treatment with LPS), especially since the bromodomain dissociation kinetics of (-)-JQ1 is also slower than that of (+)-JQ1 [50]. The MD simulation results may thus allow BET-mediated actions by (-)-JQ1, depending on experimental conditions. Again, the drug-induced changes in acetyl histone levels at the promoters did not correlate with gene activity.

In summary, our results demonstrate that BET proteins are critically involved in the response of DSCs to inflammatory stimulation. Using a panel of candidate genes, we show that both pro-and anti-inflammatory genes respond in a BET-dependent fashion and BET inhibitor drugs inhibit their responses. A BET-independent pathway also exists, mediating the stimulation of *TNF* expression by LPS. Analysis of acetylated histone levels at the promoters indicates that enhanced histone acetylation is not involved obligatorily in the transcriptional responses. BETs may regulate gene activity by acting at distant (e.g., enhancer) loci rather than at the proximal promoters. Nevertheless, BET inhibitors may represent a new class of pharmacological agents capable of arresting the inflammatory transformation of the decidua at labor with potential therapeutic benefit.

## Supporting information

**S1 Fig. Flow cytometry analysis of purified decidual stromal cells.** Decidual cells were isolated and purified by Percoll gradient centrifugation and depleted of CD45$^+$ leukocytes by immunomagnetic selection as described in the Materials and methods section. Cells ($5x10^5$) were suspended in 0.1 mL PBS, 1% BSA, and 1 μL of Human BD Fc Block (BD Pharmingen 564219) was added, followed by 2 μL of FITC mouse anti-human CD45 (BD Pharmingen 555482). Washed cells were stained with Propidium Iodide (1 μL of 0.5 mg/mL) and analyzed using a Fortessa (BD Biosciences) flow cytometer at recommended settings. **A**, Gating strategy to identify live single cells (FCC-A, forward scattering area; PI, Propidium Iodide). **B,** FITC fluorescence histogram of the gated events (FITC-A::CD45). No FITC-positive event was detected in the region of the expected CD45$^+$ subset.
(TIF)

**S2 Fig. Characterization of decidual stromal cells in culture.** Purified decidual stromal cells were cultured to near confluence as described in the Materials and methods section. Cultures were fixed with paraformaldehyde, permeabilized, and blocked with hydrogen peroxide, followed by antigen retrieval using standard procedures (Citric acid, pH 6.0, 95C for 15 min). Non-specific staining was blocked with 10% goat serum. Immunocytochemistry was performed using the VECTASTAIN ABC kit (PK6102, Vector Laboratories) following the manufacturer's instructions. The primary antibodies were against CD45 (DAKO IR751) or vimentin (DAKO IR630) used at 1:200 dilution. Control IgG was included in the staining kit. Cells were counterstained with hematoxylin. **A,** IgG negative control; **B,** CD45; **C:** vimentin. Brown staining indicates pervasive expression of the DSC marker vimentin (C), while staining for CD45 is absent (B). Magnification is shown under the panels. Size bars indicate 100 μm.
(TIF)

**S3 Fig. Inflammatory gene expression in primary cultures of decidual stromal cells (DSCs).** Purified stromal cells from term decidua were cultured to near confluence, and *PTGS1*, *PTGS2*, *PTGES*, *IL6*, *CXCL8/IL8*, *TNF*, *ILI0*, *and IDO1* mRNA abundance was determined by qRT-PCR relative to a spiked-in reference RNA (Alien). Values were scaled relative to the average in each experiment. Columns represent the mean + SEM, n = 5 independent experiments. The abundance of the mRNAs is significantly different from each other (p = 0.001, K-sample-equality-of-medians test).
(TIF)

**S4 Fig. Heat map grid showing the effects of LPS and BET inhibitors on inflammatory gene expression in DSCs.** Culture and treatment conditions, qRT-PCR assay details, and data processing are described in the Materials and methods. Genes are indicated on the top. The grid cells show unadjusted (P), and false discovery rate (FDR) adjusted significance values (Q) for all contrasts shown on the left. The significance threshold is 5% FDR (Q<0.05), highlighted in bold. Colors indicate an increase, decrease, or no significant difference (white) based on Q.
(TIF)

**S5 Fig. Heat map grid showing the effects of LPS and BET inhibitors on histone 3 acetylation at inflammatory gene promoters in DSCs.** Culture and treatment conditions, ChIP assay details, and data processing are described in the Materials and methods. Genes are indicated on the top. The grid cells show unadjusted (P), and false discovery rate (FDR) adjusted significance values (Q) for all contrasts shown on the left. The significance threshold is 5% FDR (Q<0.05), highlighted in bold. Colors indicate an increase, decrease, or no significant difference (white) based on Q.
(TIF)

**S6 Fig. Heat map grid showing the effects of LPS and BET inhibitors on histone 4 acetylation at inflammatory gene promoters in DSCs.** Culture and treatment conditions, ChIP assay details, and data processing are described in the Materials and methods. Genes are indicated on the top. The grid cells show unadjusted (P), and false discovery rate (FDR) adjusted significance values (Q) for all contrasts shown on the left. The significance threshold is 5% FDR (Q<0.05), highlighted in bold. Colors indicate an increase, decrease, or no significant difference (white) based on Q.
(TIF)

**S7 Fig. Grid showing the effects of LPS and BET inhibitors on BRD2 binding at inflammatory gene promoters in DSCs.** Culture and treatment conditions, ChIP assay details, and data processing are described in the Materials and methods. Genes are indicated on the top. The grid cells show unadjusted (P), and false discovery rate (FDR) adjusted significance values (Q) for all contrasts shown on the left. None of the contrasts reach the significance threshold of 5% FDR (Q<0.05).
(TIF)

**S8 Fig.** ChIP assays to detect BRD2 binding at the *IL6* (Panel A) *CXCL8/IL8* (Panel B) and *TNF* (Panel C) promoters in DSCs treated with LPS, (+)-JQ1 and (-)-JQ1. Primary cultures of DSCs were treated with vehicle, LPS, (+)-JQ1, (-)-JQ1, and their combinations indicated under the columns. Chromatin immunoprecipitation (ChIP) was performed using a ChIP-certified BRD2 antibody (S2 Table), and the results were expressed relative to the negative control IgG. Columns show the mean ± SEM, n = 5 independent experiments. No significant BRD2 binding relative to IgG was detected (at 5% FDR) under any treatment condition. The results

of the individual experiments and summary statistics are presented in the S1 Data.
(TIF)

**S9 Fig.** ChIP assays to detect BRD2 binding at the *IL10* (Panel A) and *IDO1* (Panel B) promoters in DSCs treated with LPS, (+)-JQ1 and (-)-JQ1. Primary cultures of DSCs were treated with vehicle, LPS, (+)-JQ1, (-)-JQ1, and their combinations indicated under the columns. Chromatin immunoprecipitation (ChIP) was performed using a ChIP-certified BRD2 antibody (S2 Table), and the results were expressed relative to the negative control IgG. Columns show the mean ± SEM, n = 4–5 independent experiments. No significant BRD2 binding relative to IgG was detected (at 5% FDR) under any treatment condition. The results of the individual experiments and summary statistics are presented in the S1 Data.
(TIF)

**S10 Fig. Grid showing the effects of LPS and BET inhibitors on BRD4 binding at inflammatory gene promoters in DSCs.** Culture and treatment conditions, ChIP assay details, and data processing are described in the Materials and methods. Genes are indicated on the top. The grid cells show unadjusted (P), and false discovery rate (FDR) adjusted significance values (Q) for all contrasts shown on the left. None of the contrasts reach the significance threshold at 5% FDR (Q<0.05).
(TIF)

**S11 Fig.** ChIP assays to detect BRD4 binding at the *IL6* (Panel A) *CXCL8/IL8* (Panel B) and *TNF* (Panel C) promoters in DSCs treated with LPS, (+)-JQ1 and (-)-JQ1. Primary cultures of DSCs were treated with vehicle, LPS, (+)-JQ1, (-)-JQ1, and their combinations indicated under the columns. Chromatin immunoprecipitation (ChIP) was performed using a ChIP-certified BRD4 antibody (S2 Table), and the results were expressed relative to the negative control IgG. Columns show the mean ± SEM, n = 5–6 independent experiments. No significant BRD4 binding relative to IgG was detected (at 5% FDR) under any treatment condition. The results of the individual experiments and summary statistics are presented in the S1 Data.
(TIF)

**S12 Fig.** ChIP assays to detect BRD4 binding at the *IL10* (Panel A) and *IDO1* (Panel B) promoters in DSCs treated with LPS, (+)-JQ1 and (-)-JQ1. Primary cultures of DSCs were treated with vehicle, LPS, (+)-JQ1, (-)-JQ1, and their combinations indicated under the columns. Chromatin immunoprecipitation (ChIP) was performed using a ChIP-certified BRD4 antibody (S2 Table), and the results were expressed relative to the negative control IgG. Columns show the mean ± SEM, n = 5–6 independent experiments. No significant BRD4 binding relative to IgG was detected (at 5% FDR) under any treatment condition. The results of the individual experiments and summary statistics are presented in the S1 Data.
(TIF)

**S1 Table. Primer sequences.**
(DOCX)

**S2 Table. Antibodies.**
(DOCX)

**S1 Data. Minimal underlying dataset of the study.**
(XLSX)

## Acknowledgments

The Authors gratefully acknowledge our Research Nurse, Ms. Anne Wright, the John Hunter Hospital Midwifery Staff, and all tissue donors for their essential contribution to the study.

## Author Contributions

**Conceptualization:** Tamas Zakar.

**Formal analysis:** Tamas Zakar.

**Funding acquisition:** Tamas Zakar.

**Investigation:** Sandeep Ajgaonkar.

**Methodology:** Sandeep Ajgaonkar.

**Project administration:** Tamas Zakar.

**Supervision:** Jonathan J. Hirst, Mary Norris, Tamas Zakar.

**Validation:** Mary Norris.

**Visualization:** Sandeep Ajgaonkar.

**Writing – original draft:** Sandeep Ajgaonkar.

**Writing – review & editing:** Jonathan J. Hirst, Tamas Zakar.

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
