## [Decision Letter · Decision Letter 0]

28 Jun 2022

PONE-D-22-13617Regulation of inflammatory genes in decidual cells: involvement of the bromodomain and extra-terminal family proteinsPLOS ONE

Dear Dr. Dr. Zakar,

Thank you for submitting your manuscript to PLOS ONE. After careful consideration, we feel that it has merit but does not fully meet PLOS ONE’s publication criteria as it currently stands. Therefore, we invite you to submit a revised version of the manuscript that addresses the points raised during the review process.

We look forward to receiving your revised manuscript.

Kind regards,

Giovanni Camussi

Academic Editor

PLOS ONE

Journal Requirements:

Reviewers' comments:

Reviewer's Responses to Questions

**Comments to the Author**

1. Is the manuscript technically sound, and do the data support the conclusions?

Reviewer #1: Partly

Reviewer #2: Yes

2. Has the statistical analysis been performed appropriately and rigorously? 

Reviewer #1: Yes

Reviewer #2: Yes

3. Have the authors made all data underlying the findings in their manuscript fully available?

Reviewer #1: Yes

Reviewer #2: Yes

4. Is the manuscript presented in an intelligible fashion and written in standard English?

Reviewer #1: No

Reviewer #2: Yes

5. Review Comments to the Author

Reviewer #1: The study investigated the interaction between the Bromodomain and Extra-Terminal (BET) family proteins and histone-3 and -4 acetylation in the control of inflammatory genes in the decidua in late pregnancy.

They used specific BET inhibitors, performing several experiments to validate their hypothesisis.

The study is of interest; however, the organization is confusing. In the first part of the result section, data reported in different figures are described. The same was for many other results. The reader should be move up and down to find the results in the figures. The Ms must be restyled reporting data in a more readable form. In the result section the legends are included. A legend section is required.

Reviewer #2: This is an interesting anaytical in vitro study on cultures of DCD cells and their response to inflammatory stimulus. Their ressults are provocative as BETs regulate pro- and anti-inflammatory genes in the decidua, and BET inhibitors abrogate the

LPS-induced pro- and anti-inflammatory gene expression, suggesting that BRD2 and BRD4, the dominant BET proteins in DSCs, interact predominantly with genomic regulatory elements distant from the promoters of the inflammatory genes studied. More imprtantly, BET inhibitors can potentially block decidual inflammation with therapeutic benefits.

GENERAL COMMENTS

Methodology

1.the authors should provide full characterization in the text; in particular, they should show cell homogeneity (exclude othe cell contamination), with a thorugh check that the biological materail they receive are not already contaminated by LPS or micoplasma

2. Based on the present knowledge of the paracrine action by which cells act, an interesting experiment would be to check whether supernatants from LPS-stimuated cells (and controls) could minick the same effect

3. the illustration of the resultslacks synthesis: the authors should organize their results without reporting them in lab-wise report sheets: a summary scheme would be beneficial

6. PLOS authors have the option to publish the peer review history of their article (what does this mean?). If published, this will include your full peer review and any attached files.

Reviewer #1: No

Reviewer #2: No

---

## [Author Response · Author response to Decision Letter 0]

20 Nov 2022

Responses to comments by the Academic Editor:

1. We have used the PLOS ONE Style templates for the formatting of the Manuscript as required. We have included a legend section, as requested by Reviewer 1. We have reformatted Figs. 2, 3, 4, 5, 7, 8, 10, 11 following the request by Dr. Maidelyn R. Peregrin from PLOS ONE.

2. We have corrected the grant information in the 'Funding Information' and the 'Financial Disclosure' sections.

3. The study's minimal underlying data set is provided as the Supporting Information File 'S1 Data'. We are ready to share all data generated in the study without restriction.

Responses to Review Comments to the Author

Comments by Reviewer #1: "...the organization is confusing". "The Ms must be restyled reporting data in a more readable form." "A legend section is required."

Responses: We have reorganized the manuscript and restyled the data presented as follows:

 We have created heat map grids that condense the effects of LPS and BET inhibitor treatments on the expression of target genes. The heat maps facilitate the summary evaluation of response patterns throughout treatments and genes. Significant stimulations and inhibitions are highlighted by colors. Fig. 1 focuses on LPS effects and BET inhibitor effects on basal expressions and LPS effects in BET inhibitor-treated cells. A heat map including all contrasts is presented in S4 Fig. for a fully detailed, comprehensive view.

 The diagrams displaying mRNA relative abundance data individually for each target gene were grouped into composite figures to reduce the need to navigate within the document. Fig. 2 includes PTGS1, PTGES, and PTGS2 of the prostaglandin synthesis pathway, Fig. 3 includes IL6, CXCL8/IL8, and TNF encoding proinflammatory factors, and Fig. 4 includes IL10, and IDO1, which are considered anti-inflammatory factors.

We have also created heat map grids to summarise the effects of LPS and BET inhibitor treatments on acH3 (Fig. 6) and acH4 (Fig. 9) levels at the gene promoters using a similar approach as in Fig. 1. The complete lists of contrasts are presented in S5 Fig., and S6 Fig. Diagrams displaying ChIP signals individually for each target gene were grouped into Figs. 7 and 8 (acH3) and Figs. 10 and 11 (acH4).

Grids and composite figures containing the BRD2 and BRD4 ChIP data are presented in the supplementary dataset since no significant effects were found (S7 - S12 Figs.).

 A legend section is included in the revised manuscript.

Comments by Reviewer #2: 

1. "...they should show cell homogeneity (exclude other cell contamination)..."

 "... a thorough check that the biological material they receive are not already contaminated by LPS or mycoplasma..."

Response: We have examined the homogeneity of the purified decidual cells by flow cytometry. The results are presented in S1 Fig. The flow cytometry analysis shows that the freshly isolated decidual cell population is not contaminated by leukocytes. 

 The cultured decidual cells have been characterized by immunocytochemistry. The results, presented in S2 Fig., demonstrate that the cells pervasively express vimentin, which is an established marker of decidual stromal cells, and do not express CD45, which is a leukocyte common antigen. These results are included in the revised text (Lines 137-139 and 143-145).

 We agree with the Reviewer that primary decidual cells may be exposed to microorganisms in situ before isolation. We have minimized this risk by collecting decidua tissue from patients without symptoms of intrauterine infection or inflammation and without signs of histological chorioamnionitis. Furthermore, the response of the cultured cells to LPS was robust, arguing against residual LPS stimulation in situ. Nevertheless, potential in situ infectious exposure represents a limitation of the study, which we acknowledge in the revised manuscript (Lines 104-105 and 380-384). 

2. "...an interesting experiment would be to check whether supernatants from LPS-stimulated cells (and controls) could mimic the same effect...

Response: We agree with the Reviewer that LPS-induced factors may be secreted in the media, potentially mediating or mimicking the effects we report. Identifying the contribution of such factors would be very important. The problem we face here is that the conditioned media still contain LPS requiring additional procedures to differentiate between bona fide LPS actions and LPS-induced mediator effects. One future research direction is to establish assay systems where bona fide LPS and LPS-induced mediator actions are discerned and characterized separately in the conditioned media.

3. "...a summary scheme would be beneficial"

Response: The heat map grids described in the responses to Reviewer #1 were designed to serve as summary schemes.

---

## [Decision Letter · Decision Letter 1]

5 Jan 2023

Regulation of inflammatory genes in decidual cells: involvement of the bromodomain and extra-terminal family proteins

PONE-D-22-13617R1

Dear Dr. Zakar,

We’re pleased to inform you that your manuscript has been judged scientifically suitable for publication and will be formally accepted for publication once it meets all outstanding technical requirements.

Kind regards,

Giovanni Camussi

Academic Editor

PLOS ONE

Additional Editor Comments (optional):

Reviewers' comments:

Reviewer's Responses to Questions

**Comments to the Author**

1. If the authors have adequately addressed your comments raised in a previous round of review and you feel that this manuscript is now acceptable for publication, you may indicate that here to bypass the “Comments to the Author” section, enter your conflict of interest statement in the “Confidential to Editor” section, and submit your "Accept" recommendation.

Reviewer #1: All comments have been addressed

Reviewer #2: All comments have been addressed

2. Is the manuscript technically sound, and do the data support the conclusions?

Reviewer #1: Yes

Reviewer #2: Yes

3. Has the statistical analysis been performed appropriately and rigorously? 

Reviewer #1: Yes

Reviewer #2: Yes

4. Have the authors made all data underlying the findings in their manuscript fully available?

Reviewer #1: Yes

Reviewer #2: Yes

5. Is the manuscript presented in an intelligible fashion and written in standard English?

Reviewer #1: Yes

Reviewer #2: Yes

6. Review Comments to the Author

Reviewer #1: Thanks to the revision the paper has been improved and more readable.

My only comment relies on the acronym description which must be reported when they are firstly described instead of in the discussion section.

Reviewer #2: The authors have fully answered to my queries and have significantly improved the quality of their manuscript

7. PLOS authors have the option to publish the peer review history of their article (what does this mean?). If published, this will include your full peer review and any attached files.

Reviewer #1: No

Reviewer #2: **Yes: **Tetta C. MD

---

## [Editor Report · Acceptance letter]

10 Jan 2023

PONE-D-22-13617R1 

Regulation of inflammatory genes in decidual cells: involvement of the bromodomain and extra-terminal family proteins 

Dear Dr. Zakar:

I'm pleased to inform you that your manuscript has been deemed suitable for publication in PLOS ONE. Congratulations! Your manuscript is now with our production department. 

Kind regards, 

on behalf of

Dr. Giovanni Camussi 

Academic Editor

PLOS ONE